# Genetic Characterization of *Salmonella* Infantis with Multiple Drug Resistance Profiles Isolated from a Poultry-Farm in Chile

**DOI:** 10.3390/microorganisms9112370

**Published:** 2021-11-17

**Authors:** Coral Pardo-Esté, Diego Lorca, Juan Castro-Severyn, Gabriel Krüger, Luis Alvarez-Thon, Phillippi Zepeda, Yoelvis Sulbaran-Bracho, Alejandro Hidalgo, Mario Tello, Franck Molina, Laurence Molina, Francisco Remonsellez, Eduardo Castro-Nallar, Claudia Saavedra

**Affiliations:** 1Laboratorio de Microbiología Molecular, Facultad de Ciencias de la Vida, Universidad Andres Bello, Santiago 8370186, Chile; cpardoeste@gmail.com (C.P.-E.); dieego.lorca@gmail.com (D.L.); g.krugercarrasco@gmail.com (G.K.); phillippi.zeps@gmail.com (P.Z.); yoelvissulbaran@gmail.com (Y.S.-B.); 2Laboratorio de Microbiología Aplicada y Extremófilos, Departamento de Ingeniería Química, Universidad Católica del Norte, Antofagasta 1240000, Chile; jsevereyn@gmail.com (J.C.-S.); fremonse@ucn.cl (F.R.); 3Facultad de Ingeniería y Arquitectura, Universidad Central de Chile, Santa Isabel 1186, Santiago 8330601, Chile; luis.alvarez@ucentral.cl; 4Escuela de Química y Farmacia, Facultad de Medicina, Universidad Andres Bello, Santiago 8370071, Chile; alejandro.hidalgo@unab.cl; 5Laboratorio de Metagenomica Bacteriana, Centro de Biotecnología Acuicola, Universidad de Santiago, Alameda, Estación Central, Santiago 9170002, Chile; mario.tello@usach.cl; 6Sys2Diag, UMR9005 CNRS ALCEDIAG, 34184 Montpellier, France; franck.molina@sys2diag.cnrs.fr (F.M.); laurence.molina@sys2diag.cnrs.fr (L.M.); 7Centro de Investigación Tecnológica del Agua en el Desierto (CEITSAZA), Universidad Católica del Norte, Antofagasta 1240000, Chile; 8Center for Bioinformatics and Integrative Biology, Facultad de Ciencias de la Vida, Universidad Andres Bello, Santiago 8370186, Chile; eduardo.castro@unab.cl; 9CIBIO-InBIO, Centro de Investigação em Biodiversidade e Recursos Genéticos, Universidade do Porto, Campus Agrário de Vairão, 4485-661 Vairão, Portugal

**Keywords:** *Salmonella* Infantis, genomic, poultry, stress resistance, virulence

## Abstract

*Salmonella* comprises over 2500 serotypes and foodborne contamination associated with this pathogen remains an important health concern worldwide. During the last decade, a shift in serotype prevalence has occurred as traditionally less prevalent serotypes are increasing in frequency of infections, especially those related to poultry meat contamination. *S.* Infantis is one of the major emerging serotypes, and these strains commonly display antimicrobial resistance and can persist despite cleaning protocols. Thus, this work aimed to isolate *S*. Infantis strains from a poultry meat farm in Santiago, Chile and to characterize genetic variations present in them. We determined their genomic and phenotypic profiles at different points along the production line. The results indicate that the strains encompass 853 polymorphic sites (core-SNPs) with isolates differing from one another by 0–347 core SNPs, suggesting variation among them; however, we found discrete correlations with the source of the sample in the production line. Furthermore, the pan-genome was composed of 4854 total gene clusters of which 2618 (53.9%) corresponds to the core-genome and only 181 (3.7%) are unique genes (those present in one particular strain). This preliminary analysis will enrich the surveillance of *Salmonella*, yet further studies are required to assess their evolution and phylogeny.

## 1. Introduction

*Salmonella enterica* is an important etiologic agent of gastroenteritis and enteric fever in a variety of hosts [1]. The disease produced by *Salmonella* infection, known as salmonellosis, remains one of the most recurrent foodborne zoonosis. Increasingly, it is a significant global health and economic problem, with thousands of cases of severe illness and deaths associated with this pathogen [2]. The food industry is constantly affected by bacterial contamination of products for human consumption, resulting in costs for surveillance prevention, disease treatment, and the loss of contaminated products [3,4]. Food-processing facilities, particularly those specialized in poultry, have a central role in the spreading of *Salmonella*. The current standard production practices such as high stocking density, larger farms, and stress result in increased occurrence and persistence of bacterial pathogens in flocks, where the main reservoir is the host’s gastrointestinal tract that can, in turn, contaminate a variety of food products [5,6]. 

Poultry, an abundant source of protein, is a primary focus of foodborne infections in humans [7,8], with *Salmonella enterica* as one of the most common pathogens associated with meat contamination. *S*. Infantis is an emerging non-typhoidal serotype associated with poultry meat and is responsible for several infection outbreaks, both in poultry and humans [4,9]. In this regard, *S.* Infantis is thus becoming a major public concern with increasing prevalence in poultry-associated infections in relation to typical serovars such as Typhimurium, Enteritidis and Gallinarium [10] as *S.* Infantis is currently among the top 10 human-associated serovars of *Salmonella* reported in several countries including Chile, and is now a major zoonosis [4,11].

Moreover, the indiscriminate use of antimicrobials has caused the spread of strains with multidrug-resistant (MDR) profiles that occur with increasing frequency [12]; the presence of MDR bacteria is a threat that could produce an epidemiological crisis. The MDR phenotype is widespread in various *Salmonella* serovars including those infecting farm animals and humans, thus presenting a challenge in current first-line therapies [13,14,15]. Clonal dissemination of MDR clones is a significant concern [16,17]. However, initial clonal strains might diversify by incorporating various genetic elements that can be incorporated into the bacterial chromosome or remain episomal, such as the pESI plasmid [18]. Additionally, recent findings using whole-genome sequencing revealed that vertical transmission is a relevant way of contamination with *S.* Enteritidis within the industry (from breeding chicken to commercial product) [19].

MDR *S.* Infantis have been isolated from different animal sources [20,21,22], increasing the risk of severe illness and highlighting the failure of standard antimicrobial chemotherapy [23]. Previously, other researchers have isolated bacteria identified as *S.* Infantis with MDR profiles, demonstrating that this phenomenon is geographically widespread with cases reported, but not limited to, Italy, Ecuador, Russia, Israel, Turkey, and the United Kingdom [15,24,25,26,27,28,29]. Furthermore, evidence supports the notion that in addition to MDR, the persistence of *Salmonella* strains throughout the production line can be associated with increased capacity to form biofilms, as an additional resistance mechanism to disinfectant agents, and the acquisition of genes involved in the global stress response [24,30,31,32,33,34].

The poultry industry implements rigorous disinfection protocols in meat production lines, mainly using 2–3% NaOCl as a microbicide compound. However, the constant use of these chemical solutions triggers tolerance, adaptation, and resistance to these agents [35,36]. Therefore, the continuous sanitization of production lines might actually enhance bacterial pathogenicity and should therefore be a major concern for the food industry and health officials, especially considering the current scenario of shifted serovar prevalence and emergence of MDR strains [37,38,39]. In this context, from 2014 to 2018, the number of clinical cases associated with *S*. Infantis strains incremented in Chile [40], and were usually associated with MDR. Poultry farms have reported the appearance of less common serotypes, that can persist, and resist disinfection protocols and regulations established by the health and food safety authorities [10,41]. This phenomenon correlates with the evolution and adaptation of specific clones that gain survival advantages through mobile elements, horizontal gene transfer and the acquisition of plasmids and prophages [29,42,43].

The bacterial mechanisms used to survive and persist along the production line include genetic traits like antibiotic resistance and virulence genes that allow the bacteria to resist the overall stress generated on the bacterial cells caused by microbicides (resistance to reactive oxygen species, rapid transcriptional response, etc.). Another reported mechanism of defense is the CRISPR-Cas system, that provides acquired immunity against viruses by targeting nucleic acids in a sequence-specific manner [44]. Additionally, horizontal gene transfer and acquisition of episomal components and other mobile genetic elements can be used as predictors of the virulence, fitness, and ability to persist of a specific strain. This information can assist the epidemiological surveillance of pathogens in the food industry, clinical settings, and in the environment [45].

In this context, we aimed to identify, characterize, and compare strains of *S.* Infantis obtained in different stages of a production line in a poultry meat production facility in the Santiago Metropolitan Region in Chile. Bacterial strains were sampled and isolated in 2018–2019 and the *S*. Infantis strains were phenotypically characterized by determining their susceptibility to antimicrobials agents (antibiotics and hypochlorous acid). Additionally, we described the genetic differences in the accessory genome in the different isolates of *S*. Infantis, using whole-genome sequencing (WGS) and bioinformatics analysis.

## 2. Materials and Methods

### 2.1. Sampling and Salmonella Isolation

Samples were taken from three stages of a chicken meat production facility in the Greater Santiago Metropolitan Area in Chile in 2018–2019. Sampled stages included: the chicken feed manufacturing, the poultry farm, and the slaughterhouse (Figure 1). Moreover, (1) the samples in the feed manufacturing stage were (I) the bran crop used to prepare the chicken feed and (II) the finished feed in the form of pellets. (2) The samples in the poultry farm stage were (III) swabs taken from the garments of farm personnel and (IV) the collected washings and swabs from the production line. (3) The samples in the slaughterhouse stage were (V) the cecum, entrails, and breast of slaughtered chickens and (VI) the collected washings from the slaughtering surfaces. We followed sample processing and isolation of *Salmonella* strains guidelines according to the ISO 6579:2002 (E) instructive. *Salmonella* strains were isolated on *Salmonella-Shigella* Agar plates (BBL™ Sparks, USA), and colorless colonies with the characteristic black center were taken and stored at −20 °C in 25% glycerol until use. We used the Check & Trace *Salmonella* rapid genomic kit (Check-Points BC^TM^) for serotyping each strain by using specific markers to identify the corresponding *Salmonella* serotype.

### 2.2. Antimicrobial Testing

We determined the antimicrobial susceptibility of the strains following the Kirby–Bauer assay by measuring the diameter of inhibition halos formed in response to antibiotic Sensi-Discs (OXOID, Thermo Scientific^®,^ Waltham, MA, USA): ampicillin (AMP), streptomycin (STR), azithromycin (AZM), nalidixic acid (NAL), tetracycline (TET), chloramphenicol (CHL), kanamycin (KAN), gentamicin (GEN), sulfamethoxazole/trimethoprim (SXT), amoxicillin-clavulanic acid (AMC), ceftriaxone (CRO), ciprofloxacin (CIP), and amikacin (AMK). Results were analyzed using the standards and cut-off values established by the Clinical and Laboratory Standards Institute (CLSI) in 2019. The *Salmonella* Typhimurium 14028s strain was used as control.

### 2.3. NaOCl Resistance

Resistance to NaOCl of all *S*. Infantis strains was assessed by determining the half-maximal inhibitory concentration (IC_50_). Briefly, bacterial cultures in Luria–Bertani broth (LB) were grown at 37 °C with aeration (150 rpm) to OD_600_ = 0.4. For each strain, a microplate with doubling serial dilutions of NaOCl in LB was set up with final concentrations ranging from 0.1 to 16 mM. To each NaOCl concentration, a 1:100 inoculum of the corresponding bacteria was added. Every strain was assayed in at least three independent experiments with six technical replicates each. Finally, plates were incubated at 37 °C for 18 h with constant agitation, and the OD_600_ was measured with an Infinite 200 PRO NanoQuant plate reader (Tecan Group Ltd., Zürich, Switzerland). The *Salmonella* Typhimurium 14028s strain was used as reference.

### 2.4. Genome Sequencing

Genomic DNA of selected *S.* Infantis strains was purified using the DNA GeneJET kit (Thermo Fisher Scientific, Waltham, MA, USA). Next, DNA integrity and concentration were determined by 1% gel electrophoresis and OD_260/280_ ratio spectroscopy. DNA samples were sent to MicrobesNG (University of Birmingham, Birmingham, UK) for paired-end libraries construction (2 × 250 bp paired-end reads: according to the manufacturer’s instructions) and sequenced on a Hiseq2500 platform (Illumina Inc., San Diego, CA, USA). Data quality was checked using FastQC v0.11.8 [46] and filtered/trimmed using PrinSeq v0.20.4 [47] (Thresholds: Ns = 0, read length ≥ 150 bp and Q ≥ 20). Read assembly was performed with SPAdes v3.7 [48] using default settings, contigs quality was checked using QUAST v5.0.2 [49] and sequencing depth coverage was calculated by mapping the reads back to the assembled contigs with BWA-MEM v0.17.7 [50]. Gene prediction and functional annotation were carried out with Prokka v1.13.3 [51] and eggNOG-mapper v1.0.3 [52] using the EggNOG v5.0 database [53]. Completion was evaluated by the search of conserved, lineage-specific orthologs with BUSCO v3 [54] and the OrthoDB v9 database [55]. The genome assemblies generated in this research have been deposited at the DDBJ/ENA/GenBank under the Bioproject: PRJNA681176.

### 2.5. Genomic Analysis

In silico Serotyping: The serotype of the sequenced strains was reassessed by querying the trimmed reads against a curated databases of *Salmonella* serotype determinants (*rfb* gene cluster, *fliC* and *fljB* alleles) as implemented in the SeqSero v1.0 tool [56]. Moreover, we performed a seven-gene (*aroC*, *dnaN*, *hemD*, *hisD*, *purE*, *sucA* and *thrA*) multi-locus sequence typing (MLST) in silico for all the strains using mlst v2.9 [57] with the pubMSLT database [58].

Genome similarities: The average nucleotide identity (ANI) matrix was calculated using pyANI v0.2.10 [59], and was used to carry out a multi-dimensional scaling (MDS) analysis to evaluate possible relationships or patterns between the genomes using R 4.0.3 [60] with the packages stats v4.0.3 and dplyr v1.0.7 (functions: dist, cmdscale and as_tibble); visualization was generated with the ggplot2 v3.3.5 [61] R package. Additionally, genome pairwise distances of the strains were calculated with Mash toolkit v2.3 [62], an “alignment-free” method which uses MinHash dimensionality-reduction technique [63] to directly estimate the nucleotide distances between two sequences. Visualization was generated with the pHeatMap v1.0.12 R package [64].

Phylogeny: A core SNP alignment was constructed using Snippy v4.3.6 [65], which identifies the core-SNPs positions (those present in every analyzed genome) against a reference genome (using default parameters and the snippy-clean_full_aln function). These core SNPs were then extracted using SNPs-sites v2.5.1 [66] (using the “-c” option). Next, a maximum likelihood phylogeny was reconstructed using the core SNPs alignment as input through the implementation of RAxML v8 [67]; with Jukes–Cantor model and Lewis ascertainment bias correction (as discussed in [68] for SNPs phylogenies). Additionally, 100,000 bootstrap and 100 searches for the best tree were performed (“-m ASC_GTRGAMMA”, “--JC69”, “--asc-corr = lewis” options). The resulting tree was visualized with the anvi’o v7 interactive interface [69]. Phylogenetic tree annotation was based on the production stage and sample type. The genome of *Salmonella* Infantis CVM-N17S1509 (GenBank: CP052817.1) was used as reference. Additionally, to calculate the pairwise distances in the number of SNPs between the genomes, we used the dist.gene function in the ape v5.5 R package [70] starting with the Snippy core-alignment. Visualization was generated with the pHeatMap v1.0.12 R package [64].

Pan-genome: The analysis was carried out following the anvi’o pangenomic workflow [69,71]. All contigs with less than 200 nucleotides were eliminated from each genome to be analyzed. Subsequently, an anvi’o genome database was generated (‘anvi-gen-genomes-storage’) to store DNA and amino acid sequences using the ‘--internal-genomes’ flag, as well as functional annotations of each gene in genomes under consideration, using HMMER [72] and COGs [73]. Next, the pangenome (‘anvi-pan-genome’) was computed from the genome database which identifies ‘gene clusters’ using Blast [74], considering only complete gene calls and ‘--minbit 0.5′ [75] and ‘--mcl-inflation 10′ [76] parameters, to remove weak hits and to identify gene clusters in the remaining blastp search results. Together, these parameters compute the occurrence of gene clusters across genomes and the total number of genes contained in each cluster. Additionally, hierarchical clustering analyses for gene clusters (based on their distribution) and for genomes (based on the gene clusters they share) were performed using Euclidean distance and Ward clustering; hence, generating a comprehensive anvi’o Pan-DataBase that stores all results for downstream analyses and visualization.

Complementary, information regarding the samples was also included in the Pan-DataBase (‘anvi-import-misc-data’), to be used to categorize genomes according to their origin. Additionally, the Average Nucleotide Identity (ANI) of each genome was computed and added to the pan database (‘anvi-compute-genome-similarity’), which uses pyANI [59]. Finally, the pangenomes were displayed (‘anvi-display-pan’) to visualize the distribution of gene clusters across genomes and interactively bin gene clusters into groups. On the other hand, we extracted unique gene clusters from the Pan-DataBase (‘anvi-get-sequences-for-gene-clusters’) for further analysis. A “gene cluster” represents sequences of one or more predicted ORFs grouped based on their homology; clusters with more than one sequence might contain orthologous, paralogous sequences or both, from one or more genomes analyzed in the pangenome.

Genes and elements search: Antimicrobial resistance (AMR) and virulence elements were detected with ABRicate v0.8.10 [77] tool using different databases: ARG-ANNOT v3 [78] for antibiotics, the Virulence Factor DataBase (VFDB) v5.0 for virulence elements [79] and PlasmidFinder v2.1 [80] for plasmid replicons (with 70% identity, 70% coverage and E-value 1 × 10^−5^ thresholds). Additionally, a search for CRISPR-Cas machinery and spacer arrays was performed using CRISPRCasTyper v1.4.1 [81] and prophage sequences within the genomes were queried with PHASTER [82].

## 3. Results

### 3.1. Serotyping and Antimicrobial Testing

To characterize the bacterial components that persist along a poultry meat production line (Figure 1), we undertook a microbiological survey and obtained isolates representative of different bacterial genera. We identified 41 *Salmonella* strains that were classified by serotyping as *Salmonella* Infantis; these bacteria were distributed in different sampling locations along the production line and were associated with specific sources (as listed in Table 1).

Isolated strains of *S.* Infantis were phenotypically characterized, focusing on relevant traits with potential to enhance pathogenesis and persistence. First, we evaluated the resistant patterns of each strain to broadly used antibiotics, discovering diversity in their antimicrobial resistance. From the analyzed strains, we found that 100% were resistant to STR, NAL and TET. Furthermore, 97% of the strains were resistant to both AZM and AMP, while 90% were resistant to KAN, 86% to GEN and STR, 83% to CHL, 28% to CRO, 17% to CXT and 14% to AMC and AMK (Figure 2). Strains SI34 and SI38 were resistant to 11 out of the 13 antimicrobials tested, representing the strains with the greatest antimicrobial resistance in this study. These strains were isolated from the slaughterhouse, present both in the poultry meat and facility surfaces. The most sensitive strains (SI01 and SI02) were isolated from the feed Pellets and could only resist 5 and 6 of the tested antimicrobials, respectively. Followed by strains SI03 and SI04 isolated from the feed manufacturing (that resist 7 antimicrobials), suggesting that the pressure of the cleaning procedures in latter stages of the production line might influence the acquisition and selection of more resistant clones.

Furthermore, the use of 2–3% of NaOCl as a disinfectant is the standard protocol used in the poultry industry; therefore, we measured the resistance of the isolated *Salmonella* strains to NaOCl and found that the IC_50_ values ranged between 2–8 mM. It is important to highlight that we found highly resistant strains in all the three stages from which samples were taken (Figure 2).

### 3.2. Genomic Features of S. Infantis Strains

Following the determination of the physiological resistance traits, we aimed to determine whether those capabilities originated from the bacterial genome. To establish specific differences in this study, we selected 29 relevant strains from the total pool based on the following criteria to cover most of the diversity: (i) most unique and contrasting strains in terms of physiological characteristics that (ii) encompassed all the production stages and sample types.

We sequenced the genomes of the 29 *S.* Infantis strains and obtained a mean of 1,003,766 reads per sample (ranging from 478,273 to 2,040,376). The average size of the genome was 4.98 Mbp (ranging from 4.96 to 5.03 Mbp). The average GC content was 52.1%, with a N50 mean value of 268,505 (ranging from 56,434 to 333,574) and the average sequencing depth coverage was 52X (ranging from 28–105X) (Table 2). Overall, the completion percentages of the genomes, provide confidence over the results. However, strains SI01 and SI02 have the least completion percentage in their genomes (<90%) and were latter removed from some analyses. Additionally, the affiliation of the genomes to *S.* Infantis was also confirmed in silico by the SeqSero tool (Appendix A), as 27 out of the 29 strains were classified as Infantis. The two remaining genomes (SI16 and SI17) were not identified as any known *Salmonella* serotype by this method. Additionally, the in silico strains typification revealed that 26 of the 29 genomes belong to the ST32 sequence type (ST), leaving SI01, SI02 and SI30 which were not assigned to any ST.

### 3.3. Genomic Similarity among the Strains

To broadly determine potential genomic changes across all isolated *S.* Infantis strains, we evaluated the average nucleotide identity of the sampled group, aiming to identify associations based on similarities between strains. However, this resulted in the absence of specific patterns and no notable groupings between the strains regarding the origin of the samples were found. Figure 3 shows the distribution of the strains according to their genome average nucleotide identity matrix, resulting in an apparent random distribution of the genomes (Figure 3a). This distribution was also confirmed according to the approximation of similarities from the Mash distances analyses (Figure 3b).

### 3.4. Phylogeny of the S. Infantis Isolates

Furthermore, we continued to characterize the phylogeny of the isolates as the broader methods (ANI and Mash) were not discriminant. We evaluated a phylogeny constructed using core SNPs and calculated the distances between the genomes using the identified core-SNPs (meaning the SNPs identified in all the analyzed strains) to determine differences as well. The SNPs were identified against the reference genome of *Salmonella* Infantis CVM-N17S1509 (GenBank: CP052817.1). For these analyzes, the strains SI01 and SI02 were not included, since they presented a low completion and do not belong to the same ST as the rest, to avoid possible pollution of the results. Hence, we analyzed their phylogenetic relationships considering 853 polymorphic sites within the core-genome (genes present in all genomes within the data set). Furthermore, we calculated pairwise distances between the genomes and found that they range varies between 0 (identical strains) to 347 (most distanced). However, we cannot conclude if these differences are due to natural selection with the disinfection protocols acting as selector or if this structure is maintained as the consequence of gene flow associated with movement of personnel or materials. Additionally, we found some isolates are phylogenetically more related, for example SI03 and IS04 both isolated from the feed manufacturing, as well as SI23 SI09 and SI10 sampled from the poultry farm suggesting that the location within the facility might be a selecting factor, however a systematic and broader study is required to confirm this (Figure 4).

### 3.5. Functional Profiles

After the previous analyses regarding serotyping, similarity and phylogenetics, we focused on specific proteins that were potentially codified by the genomes of each strain, as a means to broadly search for genetic differences. For this purpose, we classified the proteins into COG categories and compared the strains to detect differences. Overall, the functional patterns are highly convergent (Appendix A).

### 3.6. Salmonella Infantis Pangenome

To further characterize the bacterial strains and determine possible associations between them, we proceeded to analyze the pan genome—all genes and genetic variations within a given set of genomes from a species—to determine the percentage of shared and unique genes, their function, and which strains possess specific genes (Figure 5). The circular map shows the hierarchical clustering of the genomes according to the presence or absence of the 4854 identified gene clusters. Based on the lack of clustering formation between genomes by their gene patterns we can infer there are no strong associations between the strains regarding their site of isolation, suggesting gene flow and widespread distribution. On the other hand, 53.9% (2618/4854) of the identified gene clusters belong to the core-genome (in this study, we defined the core-genome as that formed by the genes found in 100% of the analyzed genomes), most of which are single-copy core genes (SCG), and only 3.7% (181/4854) are singletons or unique genes (that are present in only one genome). The remaining 42.3% (2055/4854) of the gene clusters are categorized as disposable genome, meaning the genes that are found in 2 to 26 genomes.

Furthermore, the identified 181 unique genes, were distributed between ten strains with frequencies that differ from 1 unique gene per strain to 140 unique genes per strain (SI07: 1, SI08: 1, SI34: 1, SI35: 1, SI32: 2, SI21: 8, SI26: 8, SI30: 8, SI23: 11, and SI36: 140). Most of the unique genes were found in strain SI36 that was isolated from slaughtered chicken entrails. The functions associated with the unique genes vary. Specifically, functional annotation revealed that 125/181 are hypothetical proteins (HP) that do not have an annotated known function, and 56/181 were assigned functions associated mainly with virulence and secretion systems, metabolism, protein synthesis among other key functions that might allow survival and persistence (Appendix A).

### 3.7. In Silico Study of Genetic Determinants of Resistance and Virulence Genes in S. Infantis Strains

To further characterize the strains and determine the particular traits that enabled these clones to resist standard cleaning procedures we looked specifically for differences within genes related to antimicrobial resistance and virulence. We characterized genetic traits related to antibiotic resistance (Appendix A), prophages (Appendix A), virulence-associated genes (Appendix A), CRISPR-Cas (Appendix A), and plasmids (Appendix A). In some cases, the genetic traits related to antibiotic resistance coincided with the physiological characterization (Figure 2), as it was the case with the presence of genes conferring resistance to tetracycline, aminoglycoside, and beta-lactam (Appendix A). Furthermore, we identified sequences associated with prophages within the studied strains. A total of 8 prophages were annotated; the Entero_BP_4795 was the most abundant and was found in 20 strains (Appendix A). On the other hand, plasmid IncFIB was present in all the samples included in the study; this plasmid is over 100 kb in size, and it contains many resistance cassettes and genes involved in iron acquisition [83,84]. Some strains possess two plasmids; and among them, three had the pIGMS32 plasmid (~9 kb), which contains a gene for the colicin toxin, and two had the pCROD2 (~39 kb) codifying for the toxin/antitoxin Phd/YefM (Appendix A).

A total of 117 virulence-associated genes were found, mainly related to biofilm production, pathogenicity islands, and secreted effector proteins. Some of the genes related to virulence have been previously described. These include the gene sseL, whose product is secreted during host infection to eliminate macrophages, as well as, genes related to curli fimbriae formation (csgA), necessary for invasion (invA), altering host physiology (pipB2), *Salmonella*-induced filaments formation (sifA), siderophores formation (entB), and enable ferric transport (fepG) among many others with functions that modify the host’s response to the pathogen (Appendix A) [85,86,87]. We also looked for the presence of the components of the CRISPR-Cas system. In this context, we found that all the isolated strains possess this defense strategy, varying from 2 to 3 arrays and between 46 to 53 spacers, also containing the genes cas2, cas3, cas5, cas6, and cas7 (Appendix A). Overall, our results show that the strains are distributed across all sampled sites and the abiotic pressures caused by cleaning procedures seem to select clones with MDR profiles and virulence traits that allow their survival. There is no apparent association of a specific trait with the origin of the sample, suggesting a homogeneous contamination of the food processing facility that is not sensitive to the disinfection protocols.

## 4. Discussion

In this study, we characterized *Salmonella* Infantis strains isolated from different stages along the production line of a poultry meat farm. We found that the strains share many similarities, although we were also able to find some genomic variations (Figure 6). There is no genetic structure within the sequenced *S.* Infantis among the sampled sites, the processes that are responsible for this pattern might be associated with founder effect, if all strains come from the same unsampled source. Another factor in play might be natural selection, as the facility might be seeded with different strains, but the disinfection protocols select for similar strains. Moreover, gene flow within the facility might be associated with movement of personnel and/or contaminated materials that might contribute to gene horizontal transfer or even direct contamination. In this context, there are few examples of clustering found, for instance in the poultry farm stage there are strains phylogenetically related (SI23, SI09, SI10) as well as some in the feed manufacturing (SI03 and SI04), suggesting that the procedures and stressors might to some degree be pressuring and selecting related clones, but there are also strains isolated from the same site that diverge and do not cluster. Furthermore, these strains were found on surfaces within the facility despite strict protocols of disinfection due to its high adaptability and persistence throughout the process, as was previously described [88,89]. In the isolated strains, we also determined the presence of plasmids, genes encoding for resistance and virulence factors, prophages, and CRISPR-Cas arrays that can be partly attributed to their success. This could be due to the fact that the strains acquired genomic elements and capacities, through horizontal gene transfer, that enriched the repertoire of virulence factors.

On the other hand, genome typification by MLST revealed that all the strains belong to the ST32 type, ensuring that comparisons and differences found are not due to divergences found between differences sequence types. Furthermore, in terms of phylogeny and functional properties, we determined that the isolates have hundreds of core-SNPs, suggesting the strains are diverse and the abiotic pressures might be selecting specific clones, as there is clustering between phylogenetically associated isolates, as was expected given that the stressors found along the production line is a selection factor. Additionally, the pan-genome analysis indicates that there is no association with the sample type or location. The results show a total of 4854 genes, from which 2618 (54.4%) correspond to the core-genome, leaving only 3.7% of the genes as unique.

A broad study of the geographic distribution of the Infantis serotype determined that its population evolved in three separate lineages; however, one is particularly successful in terms of infection of chickens [90]. Additionally, there are reports worldwide of isolates of *S.* Infantis with MDR, genetic homogeneity, high diversity and relevant virulence traits [10,15,25,26,37,41,89]. Here, we found high genetic diversity and widespread distribution along the production line of poultry meat with important MDR profiles and the virulence repertoire that can potentially ensure successful infection in humans.

The ability to resist antibiotics is widespread and diverse among the tested strains, and a previous study characterized *Salmonella* strains associated with broilers through pulsed field electrophoresis, and found they had high antibiotic resistance and determined three phylogenetic groups of strains [91]. MDR profiles within the Infantis serovar are common [21]; in this study, isolates with extensive multi-resistance were found in all stages of the production lines, suggesting that bacteria might be introduced by the materials used to prepare the food or other initial implements used at this stage of processing or the contamination source could be associated with tools and personnel. Additionally, this ability might be associated with the pressure and cell stress generated by standard disinfection procedures, that might trigger the acquisition or transcription of mobile and episomal factors; however, these elements might also be lost after removing the stressor in the latter stages of the process in the final product [88].

Multidrug resistant bacteria associated with human consumption products are a public health concern. In this study, we found genotypic and phenotypic evidence that this occurs in strains isolated from poultry meat. This phenomenon has been widely reported before, in Chile and in other countries [92,93,94,95,96]. The strains we isolated are resistant to commonly used antibiotics such as tetracycline, aminoglycoside, and beta-lactam. These antibiotics are extensively used in the veterinary field; therefore, their ability to resist these antimicrobials is highly prevalent in zoonotic *Salmonella* [33]. Even using technical criteria, the application of antimicrobials for treatment, prophylaxis, and as growth promoters can select resistant strains that can become problematic and of great concern for public health [97,98,99]. In Chile, a previous report detected virulent and MDR *S.* Infantis in chicken meat. In this study, the authors also highlight that environmental pollution with antibiotics in water and soil promotes the selection of multi-resistant strains [11], a factor that must be considered when formulating management and preventive strategies.

There is a strong correlation between MDR profiles and the presence of associated mobile elements, such as transmissible plasmids and the co-selection of this component because of the use of antimicrobials, producing rapid adaptation that must be challenged by effective control of zoonotic bacteria [100,101]. In this context, we identified the presence of the IncFIB plasmid in all strains, as previously reported [102], this plasmid is an important factor influencing the virulence of the strain as it carries virulence and antimicrobial resistance genes thus contributing to increased fitness [84]. Other plasmids found were pIGMS32 (~9 kb) that naturally occurs in *K. pneumoniae* [103]; and pCROD2 (~39 kb), associated with antibiotic resistance and virulence traits that would enable survival under multiple stressors by activating several bacterial response mechanisms from their repertoire.

Furthermore, as expected, all strains were able to resist high concentrations of NaOCl, the preferred microbicide used in the industry. A recent study found that the misuse of biocides triggers highly adapted bacteria that pose a critical management problem [36]. Moreover, the ability to resist high concentrations of HOCl is commonly found in strains isolated from the production line. The use of HOCl is the current standard in most disinfection protocols in the food industry [104,105], meaning that resistance traits can be found all the way to the final packed products [106]. Here, we found strains with high tolerance to this compound, that in addition to the MDR profile and other virulence traits, confer survival and persistence advantages to the bacteria.

Moreover, we found 181 unique genes, distributed among ten strains. The majority belonged to a single strain (SI36), isolated from slaughtered chicken entrails. Among the known functions associated with these unique genes are the remnants of viral genes, genes encoding virulence and secretion systems as well as metabolism- and protein synthesis-related genes. These genetic determinants might allow this strain to be highly effective in surviving the stressors found along the production line.

Pathogenicity islands (SPI) and the associated effector proteins are important traits in *Salmonella*; functions encoded in SPIs include adhesion to the host cell, membrane “ruffling” formation, and overall interference with the functioning of the eukaryotic cell preventing bacterial lysis and enabling the completion of the life cycle of *Salmonella* [85,86,87]. These are common traits in bacteria associated with food contamination and provide a great advantage to these pathogens, whilst also posing a major prophylactic and therapeutic concern that is critical in epidemiological surveillance of the food industry [11,107,108,109].

Finally, we found several genes related to the production of biofilms and fimbria, essential abilities needed to persist in the production line. Both biofilms and fimbriae are necessary for attachment and protection against microbicide compounds, like NaOCl, and allow persistence on surfaces [110,111]. Another defense mechanism present in these strains is the CRISPR-Cas system, that consists of space and repeat sequences that vary in size from 21–72 base pairs (bp) and 23–47 bp, respectively, and differ in terms of genes, number, occurrence, and size across genomes and sequences [112]. As expected, all isolated strains in this study had this defense strategy, given that it is widely spread among bacteria [44], varying between 46 to 53 spacers with the presence of *cas2*, *cas3*, *cas5*, *cas6* and *cas7* genes, indicating that these strains have integrated these components after viral infection and are now equipped for surviving viral attacks through several pathways [113].

In this study, we genotypically and phenotypically characterized strains of *Salmonella* Infantis isolated from the production line of a poultry processing facility and determined that these strains are adapted to resist all cleaning and disinfection procedures (Figure 6). The facility where this study was carried out complies with all current health and safety regulations enforced by national agencies and has also implemented a bacterial genomic surveillance strategy that provided the strains used in this study. Therefore, we hypothesized that the strains have acquired the ability to survive and persist in the production line as it obtained several genetic components, although further studies are necessary to prove the evolution and persistence of these specific clones in a long period of time including systematic sampling of all stages of production and a wider number of samples per site. Nonetheless, at the genetic level, there are few associations between a specific location or source with a particular strategy or mechanisms of survival. Moreover, the epidemiology of *Salmonella* Infantis requires more detailed analysis including monitoring for an extended period of time to address microevolution and determine if there is adaptation of a single clone or if the selective pressures drive the same characteristics into the bacterial strains. Here, we present a preliminary survey that should also be widened with studies in other facilities. We also highlight the importance of surveillance of food and products for consumption, especially regarding the MDR profiles and virulence traits that represent an imminent threat to the industry, regulators, and consumers.

## 5. Conclusions

Genomic comparisons of *Salmonella* Infantis isolates suggest a diverse group of *Salmonella* Infantis strains that are able to persist in the poultry meat production line, that survived all standard cleaning procedures. The strains identified might have acquired additional virulence traits through horizontal gene transfer along the production line. Our results indicate that current cleaning and disinfection protocols are not entirely efficient in eliminating pathogen strains of concern that cause significant health and economic issues as there is a widespread distribution of this pathogen throughout the facility. Epidemiological and surveillance of emerging *Salmonella* Infantis must therefore remain a high priority for the food industry and the governmental institutions that oversee the compliance of regulations for food safety.

## Figures and Tables

**Figure 1 microorganisms-09-02370-f001:**
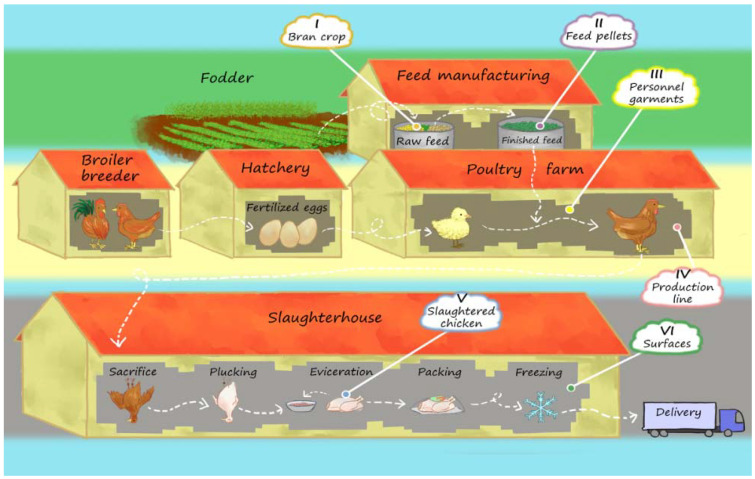
Schematic representation of the chicken meat production facility where the microbiological samples were taken. The sample types (I–VI) are pointed out in the corresponding production stages. The scheme was created by the illustrator Florence Gutzwiller (IG: @spideryscrawl).

**Figure 2 microorganisms-09-02370-f002:**
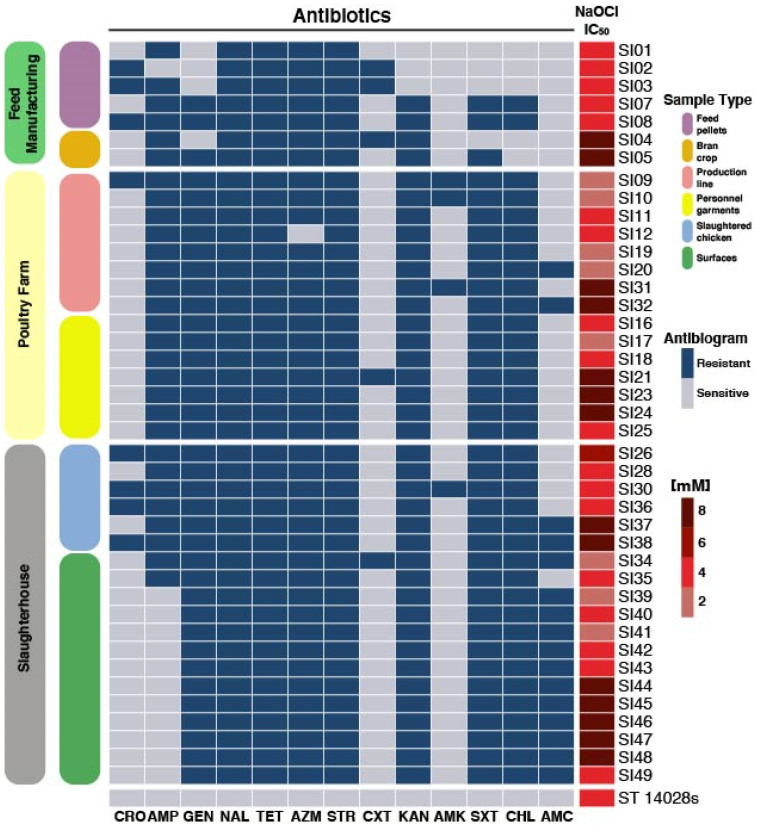
Antibiotic resistance profile of each *S.* Infantis strain (SI01–SI19) tested (AMP: ampicillin, STR: streptomycin, NAL: nalidixic acid, TET: tetracycline, AZM: azithromycin, CHL: chloramphenicol, KAN: kanamycin, GEN: gentamicin, SXT: sulfamethoxazole/trimethoprim, AMC: amoxicillin-clavulanic acid, CXT: ceftriaxone, CRO: ciprofloxacin, AMK: amikacin) according to their production stages origin. The IC_50_ value to NaOCl is also shown for each strain. The *Salmonella* Typhimurium 1028s strain (ST 14028s) was used as reference.

**Figure 3 microorganisms-09-02370-f003:**
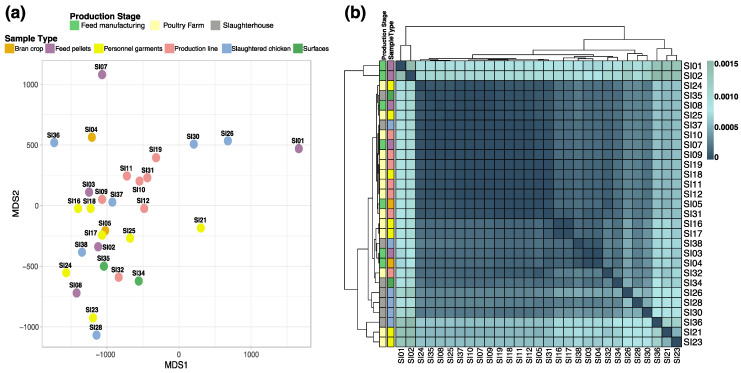
Genomic similarities among the *S.* Infantis strains. (**a**) Multidimensional scaling (MDS) plot based on average nucleotide identity (ANI) similarity index among the 29 *S.* Infantis genomes. (**b**) Heatmap representation of 29 genomes using Mash, heat scale is based on the pairwise Mash distances (identical genomes report a Mash distance of 0), the left and top dendrograms corresponds to complete-linkage hierarchical clustering. Colors of the dots and heatmap left columns represents the production stage and sample type.

**Figure 4 microorganisms-09-02370-f004:**
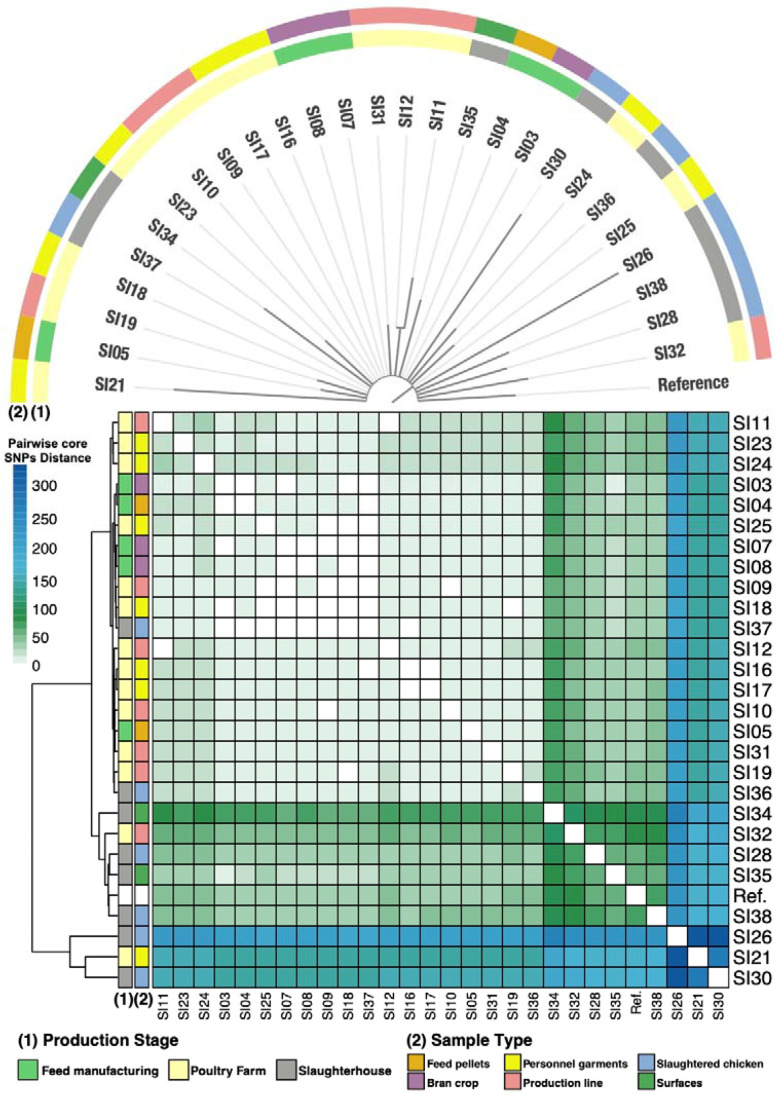
SNP-based (maximum likelihood) phylogenic tree of the 27 *S.* Infantis genomes (regarding 853 polymorphic sites across core positions). Each branch is colored according to the production stage and sample type origins. Bottom heatmap represents the pairwise distances in number of SNPs between the 27 genomes (displayed in the heat scale); identical genomes show a distance of 0. The left dendrogram corresponds to complete-linkage hierarchical clustering. Colors of the heatmap’s left columns represents the production stage and sample type. The genome of *Salmonella* Infantis CVM-N17S1509 (GenBank: CP052817.1) was used as reference.

**Figure 5 microorganisms-09-02370-f005:**
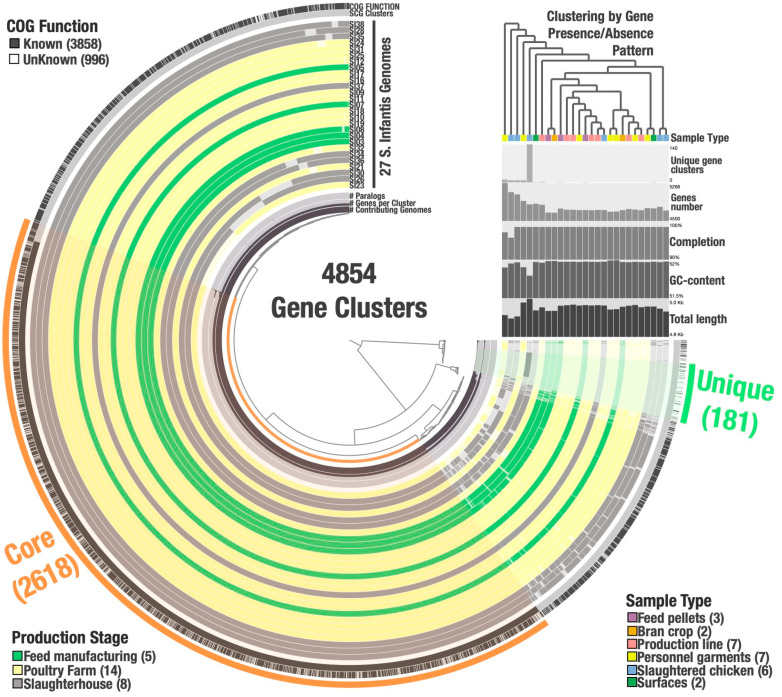
The pangenome of *S.* Infantis. Each of the 4854 gene clusters contains genes contributed by one or more isolate genomes. Gene clusters are organized based on their distribution across genomes (defined by the tree in the center), and genomes are organized in radial layers based on Euclidian distance and Ward ordination of the gene clusters they share. Visualization described from the inside out: the first three layers show (1) the number of genomes in which the corresponding cluster is present; (2) the number of genes in the corresponding cluster, and (3) the number of paralogs in the corresponding cluster. The middle 27 layers are the genomes, and the bars indicate the occurrence of a given gene cluster in that strain, colored by their origin on the production stage. The top two outside layers describe (1) the single copy core-gene clusters for the 27 genomes and (2) the gene clusters in which at least one gene was functionally annotated using COGs. Finally, the outside selections correspond to the Core gene clusters (2618), and those present exclusively in one strain (Unique: 181). The upper-right section provides additional data for each genome (from bottom to top): genome total length, GC-content (%), completion (%), number of genes, number of unique genes, the genome classification based on sample type origin and the dendrogram at the top depicts the hierarchical clustering of genomes based on the occurrence of gene clusters.

**Figure 6 microorganisms-09-02370-f006:**
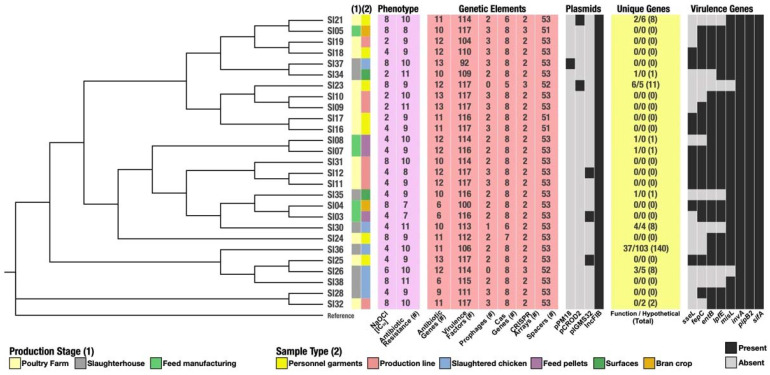
Schematic representation of the characteristics of the 27 *S*. Infantis strains. The organization is according to the SNP-based phylogeny and categorized by their production stage and sample type origins. The purple panel shows the number of antibiotics to which the corresponding strain is resistant and their IC_50_ value against NaOCl. The pink panel shows the number of different genetic elements detected in the corresponding strain. The gray heatmaps represent the presence/absence pattern for known plasmids and genes of interest (virulence). Finally, the yellow panel shows the number of unique genes: those with functional annotation, those translating to hypothetical proteins and the total of unique genes, for the corresponding strain.

**Table 1 microorganisms-09-02370-t001:** The 41 *Salmonella* Infantis strains isolated, indicating location and sample type.

Strain	Production Stage	Sample Type
SI01	Feed manufacturing	Feed pellets
SI02	Feed manufacturing	Feed pellets
SI03	Feed manufacturing	Feed pellets
SI07	Feed manufacturing	Feed pellets
SI08	Feed manufacturing	Feed pellets
SI04	Feed manufacturing	Bran crop
SI05	Feed manufacturing	Bran crop
SI09	Poultry Farm	Production line swab
SI10	Poultry Farm	Production line swab
SI11	Poultry Farm	Production line washing
SI12	Poultry Farm	Production line washing
SI19	Poultry Farm	Production line swab
SI20	Poultry Farm	Production line washing
SI31	Poultry Farm	Production line washing
SI32	Poultry Farm	Production line washing
SI16	Poultry Farm	Personnel garments swab
SI17	Poultry Farm	Personnel garments swab
SI18	Poultry Farm	Personnel garments swab
SI21	Poultry Farm	Personnel garments swab
SI23	Poultry Farm	Personnel garments swab
SI24	Poultry Farm	Personnel garments swab
SI25	Poultry Farm	Personnel garments swab
SI26	Slaughterhouse	Slaughtered chicken cecum
SI28	Slaughterhouse	Slaughtered chicken cecum
SI30	Slaughterhouse	Slaughtered chicken cecum
SI36	Slaughterhouse	Slaughtered chicken entrails
SI37	Slaughterhouse	Slaughtered chicken entrails
SI38	Slaughterhouse	Slaughtered chicken breast
SI34	Slaughterhouse	Surfaces washing
SI35	Slaughterhouse	Surfaces washing
SI39	Slaughterhouse	Surfaces washing
SI40	Slaughterhouse	Surfaces washing
SI41	Slaughterhouse	Surfaces washing
SI42	Slaughterhouse	Surfaces washing
SI43	Slaughterhouse	Surfaces washing
SI44	Slaughterhouse	Surfaces washing
SI45	Slaughterhouse	Surfaces washing
SI46	Slaughterhouse	Surfaces washing
SI47	Slaughterhouse	Surfaces washing
SI48	Slaughterhouse	Surfaces washing
SI49	Slaughterhouse	Surfaces washing

**Table 2 microorganisms-09-02370-t002:** Assembly statistics and evaluation for each of the 29 sequenced genomes of *S.* Infantis.

Sample		Strain	Size (mb)	GC (%)	# Contigs	N50	% Completion	Genome Cov.
Feed manufacturing	Feed pellets	SI01	4.96	52.15	48	333,150	88	48X
		SI02	4.95	52.14	153	56,434	77.4	37X
		SI03	4.95	52.15	65	194,600	98.4	34X
		SI07	4.98	52.14	70	148,284	99.2	32X
		SI08	4.99	52.14	49	310,053	100	44X
	Bran crop	SI04	4.95	52.15	55	204,015	99.2	63X
		SI05	4.98	52.14	54	333,144	99.2	63X
Poultry Farm	Production line	SI09	4.98	52.13	65	194,600	99.2	30X
		SI10	4.99	52.14	47	245,770	98.4	44X
		SI11	4.99	52.14	46	245,770	98.4	44X
		SI12	4.98	52.14	55	245,776	99.2	67X
		SI19	4.99	52.14	49	333,144	98.4	41X
		SI31	4.97	52.14	77	161,152	100	39X
		SI32	4.99	52.14	52	333,144	98.4	58X
	Personnelgarments swab	SI16	4.96	52.16	64	204,015	98.4	40X
		SI17	4.97	52.17	55	333,144	99.2	105X
		SI18	4.99	52.14	53	333,574	100	82X
		SI21	5.02	52.07	51	310,053	95.2	69X
		SI23	5.02	52.06	75	181,952	92.7	66X
		SI24	4.99	52.14	44	333,144	100	51X
		SI25	4.99	52.14	58	333,574	99.2	89X
Slaughterhouse	Slaughtered chicken	SI26	4.99	52.14	49	333,145	92.7	52X
		SI28	4.98	52.15	53	217,253	98.4	60X
		SI30	4.99	52.14	46	333,443	92.7	55X
		SI36	5.03	51.9	50	333,145	100	73X
		SI37	4.99	52.14	68	201,754	98.4	28X
		SI38	4.96	52.14	47	333,144	100	43X
	Surfaces	SI34	4.98	52.15	44	333,144	98.4	39X
		SI35	4.98	52.14	45	333,144	100	39X

## Data Availability

The whole raw data sets and the metagenome assembled genomes have been deposited at DDBJ/ENA/GenBank under the Bioproject: PRJNA681176.

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
