# Peer review of "Genetic Characterization of Salmonella Infantis with Multiple Drug Resistance Profiles Isolated from a Poultry-Farm in Chile"

_microorganisms, 2021, doi:10.3390/microorganisms9112370_

Round 1

Reviewer 1 Report

The authors used whole-genome sequencing to characterize 29 Salmonella Infantis strains isolated along the poultry chain. I think the study has the potential to provide an interesting and insightful look into the diversity of S. Infantis associated with poultry production operations in Chile, and I appreciate that genomic and phenotypic methods (e.g., resistance to antimicrobials) were used. However, I have some issues with the bioinformatics analyses in their current form; I do not think the major conclusion of the paper (i.e., per the abstract: “that the results suggest a common source of contamination into the production line, as there are few genomic variations between the strains”) is supported by the data the authors have collected, due to (i) potential observed genomic diversity among strains (confounded by a poor reference genome selection for SNP calling), (ii) poor genome quality, or (iii) some combination of these two scenarios. The major issues are as follows:

Reference genome selection: the authors use ANI/Mash as means to suggest that the isolates are “similar” (Lines 273-281 and Figure 3; note that methods such as ANI are usually used for applications such as species delineation and not for differentiating potentially closely related genomes), but then mention that a SNP alignment produced among the strains produced “34,055 polymorphic sites” (Line 292). These are contradictory statements; for example, in a previous study of S. Enteritidis outbreaks, S. Enteritidis genomes from the same outbreak differed from each other by <4 SNPs, while “sporadic” isolates were, on average, 42.5 SNPs distant from outbreak clusters (Taylor, et al., 2015, J Clin Microbiol). While S. Infantis is a different serotype with a different evolutionary history, an alignment with 34,055 variable sites is suspiciously large and suggests a great deal of genomic diversity among isolates.

To evaluate this, I downloaded the 35 assembled genomes associated with the BioProject accession listed in the manuscript (NCBI BioProject Accession PRJNA681176, accessed 15 October 2021), and then used strain names in Table 2 to ensure that I only included the 29 genomes discussed in the manuscript. I additionally downloaded the reference genome reportedly used for SNP calling, i.e., “the genome of Salmonella Infantis str. SARB27 (GenBank: CM001274.1)”, per Lines 185-186. Using each of the 29 assembled genomes sequenced here as input, as well as the reference genome, I performed seven-gene multi-locus sequence typing (MLST) in silico using mlst v2.9 (https://github.com/tseemann/mlst) and the seven-gene MLST scheme for Salmonella enterica (--scheme 'senterica'). Notably, the reference genome belongs to a completely different sequence type (ST) than the strains isolated in this study: reference genome CM001274.1 belongs to ST79, while 26 of 29 genomes sequenced here belong to ST32; the remaining 3 genomes sequenced here were not assigned to any ST, but their allelic profiles differed from that of ST32 by one or two allelic types (ATs). Notably, ST79 (the ST of the reference genome) and ST32 (the ST to which the majority of the isolates in the study were assigned) differ at all seven ATs, indicating that the reference genome may be comparatively distant to the genomes sequenced here.

To confirm this, I calculated average nucleotide identity (ANI) values between each of the 29 genomes sequenced here relative to the CM001274.1 reference genome via FastANI v1.31 (Jain, et al., 2018, Nature Communications). Notably, ANI values between the genomes sequenced here relative to the CM001274.1 reference genome ranged from 98.15 to 98.34 (mean and median 98.29).  For reference: some (but not all) of the 29 isolates sequenced in this study share higher ANI values (>98.70) with members of serotypes Heidelberg (NCBI SRA Accession SRR1767807), Newport (NCBI SRA Accession SRS2605991), and Paratyphi B (NCBI SRA Accession SRR2857712) than they do with the Infantis reference genome selected for SNP calling, as S. Infantis is a polyphyletic serotype (Gymoese, et al., 2019, BMC Genomics).

These results confirm that the reference genome used for SNP calling is relatively distant from the isolates sequenced in the study. This is problematic, as use of a “distant” reference genome can affect SNP identification, pairwise SNP distances between isolates, and phylogenetic topology (Olson, et al., 2015, Front Genet; Usongo, et al., 2018, PLoS One; Bush, et al., 2020, Gigascience). In Salmonella, use of a reference genome of a different serotype (which is effectively what is happening here, despite all genomes being “Infantis”, as Infantis is a polyphyletic serotype) can make isolates appear more closely related than they actually are; for example, use of an S. Dublin genome as a reference genome for SNP calling caused sporadic and outbreak S. Heidelberg isolates to incorrectly cluster together (Usongo, et al., 2018, PLoS One). Furthermore, the method used in this study for removing recombination (i.e., Gubbins) is only intended to work with intra-lineage recombination (Croucher, et al., 2014, Nucleic Acids Research), and I would be hesitant to use it across lineages, as is done here.

Genomic diversity among sequenced isolates (due to actual diversity and/or poor genome quality): I used (i) Snippy v4.4.0  (default settings; https://github.com/tseemann/snippy), (ii) Gubbins v2.4.1 (to replicate what was done in the paper submitted here, although, as mentioned above, I would not suggest using Gubbins across lineages; Croucher, et al., 2014, Nucleic Acids Research), and (iii) snp-sites v2.5.1 (Page, et al., 2016, Microbial Genomics) to identify core SNPs among the 29 isolates sequenced here, plus the reference genome used in this study (CM001274.1), using assembled genomes as input. This produced an alignment of 34,236 SNPs (very similar to what the authors report). When I calculated pairwise distances in number of SNPs (using the dist.gene function in the ape package in R v3.6.1), I find that isolates differ by 2 to 33,667 pairwise core SNPs (mean 2,292 and median 38). When I exclude the reference genome prior to running Gubbins, I find the isolates differ by 2 to 377 core SNPs (mean 66.53, median 36). These results indicate that hundreds of core SNPs may be present among the isolates sequenced here, not the “few genomic variations between strains” as described in the Abstract and elsewhere.

To ensure that the use of a relatively “distant” reference genome was not the reason for this observed “diversity”, I re-ran the SNP calling/recombination removal analysis described above, using the following as references: (i) a randomly selected closed S. Infantis ST32 chromosome (i.e., the same ST to which most strains sequenced here were assigned; NCBI RefSeq Accession GCF_000953495.1), which shared 99.79-99.89 ANI with the 29 genomes sequenced here (mean and median 99.84; calculated using FastANI as described above); and (ii) a draft genome sequenced in this study (isolate SI25, NCBI RefSeq Assembly Accession GCF_016519745.1), which shares 99.96-100.00 ANI with the 29 isolates sequenced here (mean and median 99.99). Notably for analysis (i), isolates differ by 0-449 core SNPs (mean 73.61 and median 33) when the reference is included, and by 0-383 core SNPs (mean 62.08 and median 31) when the reference is excluded. For analysis (ii), isolates differ by 1-388 core SNPs (mean 64.97 and median 33). These results indicate that, no matter which reference genome is used, some isolates sequenced in the study differ from others by hundreds of SNPs. ANI values calculated between the 29 isolates sequenced here reflect this; using FastaANI, pairwise ANI values calculated between all genomes were as low as 98.62 (as mentioned above, I would not use ANI as a method for differentiating closely related isolates; however, a value <99 is suspiciously low and suggests one or more isolates are not “closely” related and/or has poor genome quality). I would thus be hesitant to report that there are “few genomic variations between strains”, as such a large number of differences is due to (i) actual genomic diversity between strains and/or (ii) poor quality of some genomes, e.g., S01 and S02 in particular, which could also be influencing results. This also could be affecting the very small strict core genome size (see my line-by-line comments below).

References:

Bush, et al. 2020. Genomic diversity affects the accuracy of bacterial single-nucleotide polymorphism–calling pipelines. Gigascience 9(2): giaa007. https://doi.org/10.1093/gigascience/giaa007.

Croucher, et al. 2014. Rapid phylogenetic analysis of large samples of recombinant bacterial whole genome sequences using Gubbins. Nucleic Acids Research 43(3). https://doi.org/10.1093/nar/gku1196

Gymoese, et al. 2019. WGS based study of the population structure of Salmonella enterica serovar Infantis. BMC Genomics 20: 870. doi: 10.1186/s12864-019-6260-6.

Jain, et al. 2018. High throughput ANI analysis of 90K prokaryotic genomes reveals clear species boundaries. Nature Communications 9(1):5114. doi: 10.1038/s41467-018-07641-9.

Olson, et al. 2015. Best practices for evaluating single nucleotide variant calling methods for microbial genomics. Front Genet 6:235.

Page, et al. 2016. SNP-sites: rapid efficient extraction of SNPs from multi-FASTA alignments. Microbial Genomics 2(4):e000056. doi: 10.1099/mgen.0.000056.

Usongo, et al. 2018. Impact of the choice of reference genome on the ability of the core genome SNV methodology to distinguish strains of Salmonella enterica serovar Heidelberg. PLoS One https://doi.org/10.1371/journal.pone.0192233.

Additional overall notes: “Salmonella” and “S.” (e.g., in “S. Infantis”) should be italicized throughout the paper.

Line-by-line comments:

Lines 32-34: See my comments below for Lines 62-64.

Line 34: Change “serotypes” to “serotypes,”; additionally, not all S. Infantis display high resistance to antimicrobials, so I would consider re-wording.

Lines 62-64: This statement should be clarified and supported by references. Are the “infections” being referenced human infections, poultry infections, or something else? Which geographic regions/time frames is this referring to? For example, the 2018 EFSA report lists Enteritidis and Typhimurium as the two most common serotypes isolated in confirmed human salmonellosis cases, while Infantis is fourth (EFSA, 2019). The 2016 CDC report additionally has Enteritidis and Typhimurium above Infantis for culture-confirmed human Salmonella infections in the US in 2016 (CDC, 2016).

References:

EFSA. 2019. The European Union One Health 2018 Zoonoses Report. doi: 10.2903/j.efsa.2019.5926

CDC. 2016. National Enteric Disease Surveillance: Salmonella Annual Report, 2016. https://www.cdc.gov/nationalsurveillance/pdfs/2016-Salmonella-report-508.pdf

Lines 66-67: This statement should have references cited.

Lines 91-93: This statement should have references cited.

Lines 96-98: In which ways do virulence genes allow Salmonella to survive microbicides?

Line 174: Change “the pyANI” to “pyANI”; additionally, how was MDS carried out (e.g., which packages/functions were used in R, which parameters were used?)

Lines 181-182: Snippy itself does not produce a phylogeny, but rather identifies core SNPs relative to a reference genome. Consider changing “A SNP-typing phylogeny” to “A core SNP alignment”. Additionally, which reference genome was used here, and which parameters were used with Snippy? Further, I would consider re-wording Line 182, as it is difficult to understand.  

Line 183: It is unclear what “cleaned and refined” refers to here, as neither Gubbins nor snp-sites do this (Gubbins identifies/removes recombination, and snp-sites queries SNPs in an alignment). Additionally, (i) which versions of Gubbins and snp-sites were used, and (ii) what parameters were used with these tools (e.g., were SNPs produced by snp-sites core SNPs? Or something else?)

Lines 183-184: Which parameters were used to construct the phylogeny with FastTree?

Lines 185-186: Consider moving the reference genome information to the section describing SNP calling (i.e., with Snippy).

Line 214: Consider changing “assessed” to “detected”

Lines 213-216: (i) Which nucleotide identity and coverage cutoffs were used for the analyses conducted in ABRicate? (ii) What database versions were used for VFDB and PlasmidFinder (or which date were the databases accessed)?

Line 218: Which version of PHASTER was used (or, if the web tool was used, what was the date of access)?

Line 243: Change “mM, it is” to “mM. It is”

Line 247: Change “strains” to “strain”

Line 253: Change “Genomes” to “Genomic”

Line 255: Change “were originated” to “originated”

Line 258: Change “encompassing” to “encompassed”

Lines 263-267: How was (i) sequencing depth coverage and (ii) genome completion assessed (e.g., which software was used)? It may be useful to briefly specify the tools/methods used to calculate these values specifically.

Line 268: “in silico” should be italicized.

Table 2: There are many formatting characters here that make this table difficult to read (dollar signs, apostrophes, etc.)

Lines 273-281: Distance-based methods like ANI and Mash are not usually used for assessing diversity among closely related strains (e.g., see Bush, et al., 2020, Gigascience). Based on the alignment of SNPs produced using Snippy, what was the pairwise SNP range between strains (e.g., this can be calculated using the dist.gene function in the ape package in R)?

References:

Bush, et al. 2020. Genomic diversity affects the accuracy of bacterial single-nucleotide polymorphism–calling pipelines. Gigascience 9(2): giaa007. https://doi.org/10.1093/gigascience/giaa007.

Figure 3: Which method was used to construct dendrograms (e.g., complete linage, average linkage)? Additionally, what is the upper bound on the scale bar (i.e., the bar shows zero, but no maximum distance value)?

Lines 292: “34,055 polymorphic sites” is an extremely large number of SNPs, indicating that the isolates are not particularly closely related and do not showcase clonality (see my discussion above).

Lines 293-295: It is unclear what “no divergences were found associated with a specific sample site” means here.

Figure 4: How was the phylogeny rooted, and what is the scale of the branch lengths? Additionally, were any methods used to assess branch support (e.g., bootstrapping)?

Lines 312-313: 1,786 (37%) core genes is extremely low for a core genome size for Salmonella, particularly among isolates of a single serotype, and even more so among isolates that are supposedly of a single origin, as is reported throughout the paper. For example, a previous study of S. Infantis ST32 reports a strict core genome size of 3,552 genes/3,161,448 bp (Mejia, et al., 2020, Front Vet Sci). The small size of the core genome in this study could be due to either (i) diversity, which is unaccounted for among S. Infantis genomes here and/or (ii) the poor quality of some genomes included in the analysis.

References:

Mejia, et al. 2020. Genomic Epidemiology of Salmonella Infantis in Ecuador: From Poultry Farms to Human Infections. Front Vet Sci https://doi.org/10.3389/fvets.2020.547891.

Figure 5: 90 ANI is extremely low (e.g., 95 is usually a species cutoff); furthermore, it would be useful to have a guide showcasing the numerical values associated with the shading of ANI values.

Lines 363: References should be provided, which support the statement that plasmids pPM18 and Col440I plasmids are associated with enterobacteria outbreaks.

Lines 364-378: Gene names should be italicized here.

Lines 374-378, 399-404: As detailed above, I would be hesitant to suggest all strains are highly convergent on a genomic scale.

Figure 6: How was the phylogeny rooted here?

Line 431-433: It is unclear how IncFIB’s association with virulence relates to MDR and mobile genetic elements, as is discussed in the previous lines (Lines 428-431).

Lines 435-436: How would virulence traits enhance survival/persistence in the food processing environment?

Author Response

Reviewer 1

Reference genome selection: the authors use ANI/Mash as means to suggest that the isolates are “similar” (Lines 273-281 and Figure 3; note that methods such as ANI are usually used for applications such as species delineation and not for differentiating potentially closely related genomes), but then mention that a SNP alignment produced among the strains produced “34,055 polymorphic sites” (Line 292). These are contradictory statements; for example, in a previous study of S. Enteritidis outbreaks, S. Enteritidis genomes from the same outbreak differed from each other by <4 SNPs, while “sporadic” isolates were, on average, 42.5 SNPs distant from outbreak clusters (Taylor, et al., 2015, J Clin Microbiol). While S. Infantis is a different serotype with a different evolutionary history, an alignment with 34,055 variable sites is suspiciously large and suggests a great deal of genomic diversity among isolates.

To evaluate this, I downloaded the 35 assembled genomes associated with the BioProject accession listed in the manuscript (NCBI BioProject Accession PRJNA681176, accessed 15 October 2021), and then used strain names in Table 2 to ensure that I only included the 29 genomes discussed in the manuscript. I additionally downloaded the reference genome reportedly used for SNP calling, i.e., “the genome of Salmonella Infantis str. SARB27 (GenBank: CM001274.1)”, per Lines 185-186. Using each of the 29 assembled genomes sequenced here as input, as well as the reference genome, I performed seven-gene multi-locus sequence typing (MLST) in silico using mlst v2.9 (https://github.com/tseemann/mlst) and the seven-gene MLST scheme for Salmonella enterica (--scheme 'senterica'). Notably, the reference genome belongs to a completely different sequence type (ST) than the strains isolated in this study: reference genome CM001274.1 belongs to ST79, while 26 of 29 genomes sequenced here belong to ST32; the remaining 3 genomes sequenced here were not assigned to any ST, but their allelic profiles differed from that of ST32 by one or two allelic types (ATs). Notably, ST79 (the ST of the reference genome) and ST32 (the ST to which the majority of the isolates in the study were assigned) differ at all seven ATs, indicating that the reference genome may be comparatively distant to the genomes sequenced here.

To confirm this, I calculated average nucleotide identity (ANI) values between each of the 29 genomes sequenced here relative to the CM001274.1 reference genome via FastANI v1.31 (Jain, et al., 2018, Nature Communications). Notably, ANI values between the genomes sequenced here relative to the CM001274.1 reference genome ranged from 98.15 to 98.34 (mean and median 98.29).  For reference: some (but not all) of the 29 isolates sequenced in this study share higher ANI values (>98.70) with members of serotypes Heidelberg (NCBI SRA Accession SRR1767807), Newport (NCBI SRA Accession SRS2605991), and Paratyphi B (NCBI SRA Accession SRR2857712) than they do with the Infantis reference genome selected for SNP calling, as S. Infantis is a polyphyletic serotype (Gymoese, et al., 2019, BMC Genomics).

These results confirm that the reference genome used for SNP calling is relatively distant from the isolates sequenced in the study. This is problematic, as use of a “distant” reference genome can affect SNP identification, pairwise SNP distances between isolates, and phylogenetic topology (Olson, et al., 2015, Front Genet; Usongo, et al., 2018, PLoS One; Bush, et al., 2020, Gigascience). In Salmonella, use of a reference genome of a different serotype (which is effectively what is happening here, despite all genomes being “Infantis”, as Infantis is a polyphyletic serotype) can make isolates appear more closely related than they actually are; for example, use of an S. Dublin genome as a reference genome for SNP calling caused sporadic and outbreak S. Heidelberg isolates to incorrectly cluster together (Usongo, et al., 2018, PLoS One). Furthermore, the method used in this study for removing recombination (i.e., Gubbins) is only intended to work with intra-lineage recombination (Croucher, et al., 2014, Nucleic Acids Research), and I would be hesitant to use it across lineages, as is done here.

We thank the reviewer for taking the time to perform this in-depth analysis, as the evidence is clear we proceeded to change the reference genome used, choosing the genome of Salmonella Infantis CVM-N17S1509 (GenBank: CP052817.1) that is closely related to the studied strains (ANI), was isolated from chicken breast, and also belongs to the ST32. With this reference we redid the SNPs-based phylogeny and the Pan-Genome analysis, which can be seen in updated Figs 4, 5 and 6, thus adapting the results and discussion accordingly.

Genomic diversity among sequenced isolates (due to actual diversity and/or poor genome quality): I used (i) Snippy v4.4.0  (default settings; https://github.com/tseemann/snippy), (ii) Gubbins v2.4.1 (to replicate what was done in the paper submitted here, although, as mentioned above, I would not suggest using Gubbins across lineages; Croucher, et al., 2014, Nucleic Acids Research), and (iii) snp-sites v2.5.1 (Page, et al., 2016, Microbial Genomics) to identify core SNPs among the 29 isolates sequenced here, plus the reference genome used in this study (CM001274.1), using assembled genomes as input. This produced an alignment of 34,236 SNPs (very similar to what the authors report). When I calculated pairwise distances in number of SNPs (using the dist.gene function in the ape package in R v3.6.1), I find that isolates differ by 2 to 33,667 pairwise core SNPs (mean 2,292 and median 38). When I exclude the reference genome prior to running Gubbins, I find the isolates differ by 2 to 377 core SNPs (mean 66.53, median 36). These results indicate that hundreds of core SNPs may be present among the isolates sequenced here, not the “few genomic variations between strains” as described in the Abstract and elsewhere.

To ensure that the use of a relatively “distant” reference genome was not the reason for this observed “diversity”, I re-ran the SNP calling/recombination removal analysis described above, using the following as references: (i) a randomly selected closed S. Infantis ST32 chromosome (i.e., the same ST to which most strains sequenced here were assigned; NCBI RefSeq Accession GCF_000953495.1), which shared 99.79-99.89 ANI with the 29 genomes sequenced here (mean and median 99.84; calculated using FastANI as described above); and (ii) a draft genome sequenced in this study (isolate SI25, NCBI RefSeq Assembly Accession GCF_016519745.1), which shares 99.96-100.00 ANI with the 29 isolates sequenced here (mean and median 99.99). Notably for analysis (i), isolates differ by 0-449 core SNPs (mean 73.61 and median 33) when the reference is included, and by 0-383 core SNPs (mean 62.08 and median 31) when the reference is excluded. For analysis (ii), isolates differ by 1-388 core SNPs (mean 64.97 and median 33). These results indicate that, no matter which reference genome is used, some isolates sequenced in the study differ from others by hundreds of SNPs. ANI values calculated between the 29 isolates sequenced here reflect this; using FastaANI, pairwise ANI values calculated between all genomes were as low as 98.62 (as mentioned above, I would not use ANI as a method for differentiating closely related isolates; however, a value <99 is suspiciously low and suggests one or more isolates are not “closely” related and/or has poor genome quality). I would thus be hesitant to report that there are “few genomic variations between strains”, as such a large number of differences is due to (i) actual genomic diversity between strains and/or (ii) poor quality of some genomes, e.g., S01 and S02 in particular, which could also be influencing results. This also could be affecting the very small strict core genome size (see my line-by-line comments below).

We thank the reviewer for these comments, as suggested:

  • We changed the reference strain for a proper one: Salmonella Infantis CVM-N17S1509 that shares ANI values from 99.68 to 99.96% with the presented strains, belongs to the same ST32 (MLST) and was also isolated from chicken breast.
  • The new version of the SNP-based phylogeny resulted in a total of 853 polymorphic sites detected (ranging from 49 to 312 SNPs).
  • We included the Ape SNPs distance analysis: we found a range from 0 to 347 (which was included in the revised version of the manuscript as part of the new Fig 4.
  • We removed strains S01 and S02 as suggested, from the SNPs-based phylogeny and Pan-Genome analysis: indeed, those strains presented the lowest completion level (below 90% according to the BUSCO analysis).
  • The Pan-Genome proportions changed vastly with the removal of the two strains, an increase from 37.1 to 53.9 % in the core genome compartment and a reduction from 188 to 181 unique genes.

Finding changes were made throughout the manuscript (especially evidenced in Figs 4, 5 and 6).

Additional overall notes: “Salmonella” and “S.” (e.g., in “S. Infantis”) should be italicized throughout the paper.

 This seems to be a format issue, as the manuscript we provided had “Salmonella” and “S” italicized, however the PDF that the website generated does not. Nonetheless, we checked again in all instances.

Line-by-line comments:

Lines 32-34: See my comments below for Lines 62-64.

 This statement has been modified to explain the intended idea more accurately (Lines 35-36).

Line 34: Change “serotypes” to “serotype,”; additionally, not all S. Infantis display high resistance to antimicrobials, so I would consider re-wording.

We re-worded the lines regarding the bacterial resistance (Line 37). However, we maintained “serotypes” as we are referring to the plural form of the word (several serotypes that are emerging, including Infantis).

Lines 62-64: This statement should be clarified and supported by references. Are the “infections” being referenced human infections, poultry infections, or something else? Which geographic regions/time frames is this referring to? For example, the 2018 EFSA report lists Enteritidis and Typhimurium as the two most common serotypes isolated in confirmed human salmonellosis cases, while Infantis is fourth (EFSA, 2019). The 2016 CDC report additionally has Enteritidis and Typhimurium above Infantis for culture-confirmed human Salmonella infections in the US in 2016 (CDC, 2016).

This statement has been corrected (Lines 35-36, 68-71).

Lines 66-67: This statement should have references cited.

Done. Line 74

Lines 91-93: This statement should have references cited.

 Done. Line 83-83, 103

Lines 96-98: In which ways do virulence genes allow Salmonella to survive microbicides?

 This phrase was changed to better explain the idea that the presence of virulence genes that enable the bacteria to utilize difference energy sources, to effectively activate a transcriptional/expressional response, to prioritize evasion mechanisms over energy production, the secretion of extracellular components and secretions that physically and physiologically protect the cell, as well as biosignatures that indicate stress to other bacteria, and the overall stress response associated with virulence and fitness can ultimately allow Salmonella to survive microbicides and other stressors (Crane et al., 2021 PMID: 33875437; Flores-Kim et al., 2014 PMID: 25603429, for some examples). (Lines 108-110).

Line 174: Change “the pyANI” to “pyANI”; additionally, how was MDS carried out (e.g., which packages/functions were used in R, which parameters were used?)

 Done.

Lines 181-182: Snippy itself does not produce a phylogeny, but rather identifies core SNPs relative to a reference genome. Consider changing “A SNP-typing phylogeny” to “A core SNP alignment”. Additionally, which reference genome was used here, and which parameters were used with Snippy? Further, I would consider re-wording Line 182, as it is difficult to understand.  

 Agreed, clarifications have been made in this section (see lines 198-208).

Line 183: It is unclear what “cleaned and refined” refers to here, as neither Gubbins nor snp-sites do this (Gubbins identifies/removes recombination, and snp-sites queries SNPs in an alignment). Additionally, (i) which versions of Gubbins and snp-sites were used, and (ii) what parameters were used with these tools (e.g., were SNPs produced by snp-sites core SNPs? Or something else?)

 This have been rephrased in the revised version of the manuscript to clarify (see lines 198-208).

Lines 183-184: Which parameters were used to construct the phylogeny with FastTree?

In the original version we used a generalized time-reversible model, the phylogeny presented in the revised manuscript was reconstructed using Bayesian inference with MrBayes software (1 million generations were executed; sampling every 1000 and 10% of burnin).

Lines 185-186: Consider moving the reference genome information to the section describing SNP calling (i.e., with Snippy).

Done, the new reference genome used has been stated in the corresponding section (see lines 205-206).

Line 214: Consider changing “assessed” to “detected”

Done. Line 237

Lines 213-216: (i) Which nucleotide identity and coverage cutoffs were used for the analyses conducted in ABRicate? (ii) What database versions were used for VFDB and PlasmidFinder (or which date were the databases accessed)?

  • 70% coverage and 70% identity (this has been added to the revised manuscript: lines 238-240).
  • Virulence Factor DataBase (VFDB) v5.0 and and PlasmidFinder v2.1 (this information has been added to the revised manuscript, see lines 238-240.

Line 218: Which version of PHASTER was used (or, if the web tool was used, what was the date of access)?

We accessed the tool in March of 2021.

Line 243: Change “mM, it is” to “mM. It is”

Done. Line 270

Line 247: Change “strains” to “strain”

Done. Line 274

 Line 253: Change “Genomes” to “Genomic”

 Done. Line 281

Line 255: Change “were originated” to “originated”

 Done. Line 283

Line 258: Change “encompassing” to “encompassed”

Done. Line 286

 Lines 263-267: How was (i) sequencing depth coverage and (ii) genome completion assessed (e.g., which software was used)? It may be useful to briefly specify the tools/methods used to calculate these values specifically.

  • Sequencing depth coverage was calculated by mapping the reads back to the assembled contigs with BWA-MEM v0.17.7.
  • This was assessed with the BUSCO software, which looks for a set of conserved orthologs in the genome

This information has been added to the revised manuscript (see lines 176-178, 179-181)

Line 268: “in silico” should be italicized.

 Done.

Table 2: There are many formatting characters here that make this table difficult to read (dollar signs, apostrophes, etc.)

We changed the format of the table to avoid this issue.

Lines 273-281: Distance-based methods like ANI and Mash are not usually used for assessing diversity among closely related strains (e.g., see Bush, et al., 2020, Gigascience). Based on the alignment of SNPs produced using Snippy, what was the pairwise SNP range between strains (e.g., this can be calculated using the dist.gene function in the ape package in R)?

As suggested, we included the Ape SNPs distance analysis, and found a range from 0 to 347 (see revised version of the manuscript as part of the new Fig. 4); thus, adjusting the results throughout the manuscript.

Figure 3: Which method was used to construct dendrograms (e.g., complete linage, average linkage)? Additionally, what is the upper bound on the scale bar (i.e., the bar shows zero, but no maximum distance value)?

The dendrogram are complete-linkage Hierarchical Clustering between the samples (this has been added to the figure caption).

This was a formatting mistake, the upper bound of the scale is 0.0015, which was corrected in the revised version.

Lines 292: “34,055 polymorphic sites” is an extremely large number of SNPs, indicating that the isolates are not particularly closely related and do not showcase clonality (see my discussion above).

The new results obtained with the method that the review suggests indicating 853 polymorphic sites.

Lines 293-295: It is unclear what “no divergences were found associated with a specific sample site” means here.

 This whole section was revised and corrected. (Lines 329-337)

Figure 4: How was the phylogeny rooted, and what is the scale of the branch lengths? Additionally, were any methods used to assess branch support (e.g., bootstrapping)?

 A new phylogeny was calculated for the revised version. In new Fig. 4 the tree is mid-point rooted and in Fig. 6 the tree was rooted to the reference genome, also in this case the bootstrap supporting values are displayed. Lines 33-346

Lines 312-313: 1,786 (37%) core genes is extremely low for a core genome size for Salmonella, particularly among isolates of a single serotype, and even more so among isolates that are supposedly of a single origin, as is reported throughout the paper. For example, a previous study of S. Infantis ST32 reports a strict core genome size of 3,552 genes/3,161,448 bp (Mejia, et al., 2020, Front Vet Sci). The small size of the core genome in this study could be due to either (i) diversity, which is unaccounted for among S. Infantis genomes here and/or (ii) the poor quality of some genomes included in the analysis.

 In the new analyses we determined a core genome corresponding to 2,618 gene clusters (54.4%), we adapt to these new results all finding regarding the pangenome. The values changed once we removed the two low-quality genomes (SI01 and SI02). Lines 359, 360, 363, 364, 367-384.

Figure 5: 90 ANI is extremely low (e.g., 95 is usually a species cutoff); furthermore, it would be useful to have a guide showcasing the numerical values associated with the shading of ANI values.

 This was an error caused during plotting, that we didn’t noticed, not regarding the analysis results per se. Nonetheless, the ANI was removed from the new Pan-Genome figure 5, because after considering it we found it redundant.

Lines 363: References should be provided, which support the statement that plasmids pPM18 and Col440I plasmids are associated with enterobacteria outbreaks.

 This paragraph was modified to make more emphasis on plasmids found in more strains since plasmid pPM18 was only found in one strain and its association with virulence is minimal in the literature (PMCID: PMC217982), also plasmid Col4401 was found only in a strain with low quality therefore it was removed from the discussion.

Lines 364-378: Gene names should be italicized here.

 This seems to be a format issue, as the manuscript we provided had the gene names italicized, however the PDF that the website generated does not. We checked again in all instances

Lines 374-378, 399-404: As detailed above, I would be hesitant to suggest all strains are highly convergent on a genomic scale.

 This was corrected throughout the manuscript

Figure 6: How was the phylogeny rooted here?

The original tree version was rooted selecting the reference genome as “outgroup”.

Line 431-433: It is unclear how IncFIB’s association with virulence relates to MDR and mobile genetic elements, as is discussed in the previous lines (Lines 428-431).

This was addressed, reworded and we added more information to explain this relationship in Lines 503-505.

Lines 435-436: How would virulence traits enhance survival/persistence in the food processing environment?

This sentence was changed to better describe the idea, and we also eliminated plasmid that are no longer included in the study. (Lines 507-509).

Reviewer 2 Report

The manuscript entitled "Genetic characterization of Salmonella Infantis with Multiple 2 Drug Resistance profiles isolated from a poultry-farm in Chile" is an original article aiming to isolate S. Infantis strains from a poultry meat farm in Santiago, Chile and to characterize genetic variations present in them. Although it is a wonderful work and well presented, the discussion could be improved to get more merit.

Major revision:

There is a very similar study “Phenotypic and genetic characterization of multidrug-resistant Salmonella Infantis strains isolated from broiler chicken meats in Turkey” reported in 2011 by Abbasoglu D and Akcelik M (Biologia 2011). What is the difference between these two studies? Is there any dramatic trend from 2011 to 2021 we have to notice in manipulating the poultry? Additionally, there are some studies from different country reporting genetic characteristics of Salmonella Infantis. Is there any geographic diversity? The authors should discuss them in the revised manuscript.

Minor revision:

The references should be decreased. The most relevant and important references could be remained.

Author Response

Reviewer 2

Major revision:

There is a very similar study “Phenotypic and genetic characterization of multidrug-resistant Salmonella Infantis strains isolated from broiler chicken meats in Turkey” reported in 2011 by Abbasoglu D and Akcelik M (Biologia 2011). What is the difference between these two studies? Is there any dramatic trend from 2011 to 2021 we have to notice in manipulating the poultry? Additionally, there are some studies from different country reporting genetic characteristics of Salmonella Infantis. Is there any geographic diversity? The authors should discuss them in the revised manuscript.

We thank the reviewer for highlighting this aspect of the discussion, we included in Lines 465-483 references and statements addressing this. There is indeed a concerning trend of MDR profiles worldwide, that must be addressed not only in the context of zoonosis but also as the imminent thread of easily spread genes by horizontal transfer.

Minor revision:

The references should be decreased. The most relevant and important references could be remained.

We addressed this, however as new analyses were included (as requested by reviewers) we also included some references for methodological purposes and in the introduction and discussion sections. Thus, we also included more references as requested by reviews to better explain and support our findings.

Reviewer 3 Report

The manuscript reported the ggenetic characterization of Salmonella Infantis with multiple drug resistance profiles isolated from a poultry-farm in Chile. Overall, the study is of interest for readers. Some minor issues should be addressed.

Lines 28-35, Abstract: The background is too long. I suggest the authors cut down it and add more detailed results.

Lines 71-73, the article (Vet Microbiol. 2020 Jan;240:108538) also reported acquisition of antimicrobial-resistant plasmid or Salmonella virulence plasmid evolving into virulence-resistance plasmids can result in the heterogeneous antimicrobial resistance of clonally related S. Enteritidis strains from poultry.

Lines 114-122, how many samples were collected in every stages?

Line 139, Ceftriaxone, CRO?

Line 223, The authors state 41 Salmonella strains that were classified by serotyping as Salmonella Infantis. How many Salmonella strains were isolated in total? Is S. Infantis the primary serotype?

How about the sequence types (MLST) of the 29 S. Infantis strains?

Author Response

Reviewer 3

The manuscript reported the genetic characterization of Salmonella Infantis with multiple drug resistance profiles isolated from a poultry-farm in Chile. Overall, the study is of interest for readers. Some minor issues should be addressed.

Lines 28-35, Abstract: The background is too long. I suggest the authors cut down it and add more detailed results.

Done, we have revised the entire abstract. (Lines 32-45).

Lines 71-73, the article (Vet Microbiol. 2020 Jan;240:108538) also reported acquisition of antimicrobial-resistant plasmid or Salmonella virulence plasmid evolving into virulence-resistance plasmids can result in the heterogeneous antimicrobial resistance of clonally related S. Enteritidis strains from poultry.

We have included this finding in Lines 81-83.

Lines 114-122, how many samples were collected in every stages?

The facility of poultry meat processing carries out periodic bacterial samples as part of their microbiology surveillance strategy and we obtained the 61 Salmonella isolates (once identified in selective medium) from those sampling endeavors, therefore we do not have any information regarding the total amount of samples or other bacterial species collected.

Line 139, Ceftriaxone, CRO?

This was corrected. in Figure 2, Lines 152, 258 and 277

Line 223, The authors state 41 Salmonella strains that were classified by serotyping as Salmonella Infantis. How many Salmonella strains were isolated in total? Is S. Infantis the primary serotype?

We isolated 61 Salmonella strains, 20 were identified as Typhimurium and 41 as Infantis

How about the sequence types (MLST) of the 29 S. Infantis strains?

We have included this, see Figure 4 and Lines 186-189, 445-457.

Round 2

Reviewer 1 Report

I appreciate the extensive methodological changes made to improve the quality of the manuscript. However, there are some parts of the methods, as well as some results,  that are not described in enough detail. Line-by-line comments are as follows:

Line 32: This sentence sounds a little strange; possibly consider changing “spanning a great variety of hosts” to something like “capable of infecting a variety of hosts” (if possible; I understand the abstract likely has word/character limits).  

Lines 36-37: Change “emerging serotypes and these” to “emerging serotypes, and these”

Lines 41-42: Based on the responses to reviewers, I am assuming the “0-347” refers to pairwise SNP differences calculated between strains; if this is correct, I would thus re-word to something like “The results indicate that the strains encompass 853 polymorphic sites (core-SNPs), with isolates differing from one another by 0-347 core SNPs”, as “per strain” here is ambiguous.

Lines 191-192: Which R functions/packages were used to perform MDS?

Line 197: Change “with pHeatMap” to “with the pHeatMap”

Lines 199: I would suggest changing “the changes in the genome core-positions” to “core SNPs” for conciseness/readability, as the original sentence sounds strange/is difficult to follow.

Line 201: In the previous version of the manuscript, Gubbins was used to remove recombination prior to running snp-sites; why was this step omitted this time?  

Lines 201-202: I’m assuming core SNPs were extracted usng snp-sites; if this is the case, I would change “These polymorphic positions” to “Core SNPs”, and specify that the “-c” option was used with snp-sites (i.e., the option which outputs a core SNP alignment).

Lines 202-203: Many more details need to be provided regarding how MrBayes was used here, as it is not replicable as-is; e.g., I’m assuming core SNPs produced by Snippy/snp-sites were used as input; was an ascertainment bias correction applied to account for the use of solely variant sites, and if so, how was this performed? Which model was specified? Which priors were used?

Lines 205-206: I would suggest moving this sentence up to where Snippy is discussed, as the reference genome is used with Snippy/relevant for SNP calling.

Line 209: Change “with pHeatMap” to “with the pHeatMap”

Line 323: Consider changing “SNPs phylogeny” to “a phylogeny constructed using core SNPs”

Lines 331-333: What evidence is there to support the claim that slaughterhouse conditions are more likely to favor more diverse strains (e.g., how can one rule out that a strain, which is not adapted to the slaughterhouse environment, was introduced recently into the slaughterhouse from an external location/source?)?

Lines 359-360: How were associations determined (or, in this case, the lack thereof) here?  

Lines 406-408: Because the genomes here are draft genomes, one cannot prove definitively that the IncFIB plasmid harbored the AMR and/or virulence genes; was the IncFIB replicon colocalized with the AMR/virulence factors (i.e., were these elements all on the same contig)? This may offer more proof to support this statement.  

Lines 409-410: What evidence is there that these virulence factor determinants are on the specified plasmids, given that the genomes are not closed genomes (see my comment for 406-408 above)?

Line 446: Consider changing “SNPs-based” to “SNP-based”, as it sounds better.

Line 447-448: What was the rationale for rooting the tree using the reference genome? Based on the midpoint rooted phylogeny (Figure 4) and core SNP distance matrix, the reference genome doesn’t appear to be an outgroup.

Lines 445-454: What are the branch length units/tree scale? Additionally, what do the node labels of the phylogeny denote?  

Lines 457-461: This is confusing, as above (Lines 431-437) it is mentioned that there is “an apparent random distribution of the strains” and that “strong associations or clustering based on genomic variability and source of the sample within the production line”…”could not be detected”.

Line 484: Change “resistance bacteria” to “resistant bacteria”

Line 488: It is unclear why mealworms are discussed here

Supplementary Figures: The Supplementary Figures would benefit from figure legends.

Author Response

Reviewer 1:

Line 32: This sentence sounds a little strange; possibly consider changing “spanning a great variety of hosts” to something like “capable of infecting a variety of hosts” (if possible; I understand the abstract likely has word/character limits).  

This line was removed due to character limits and the prioritization of the results section of the abstract.

Lines 36-37: Change “emerging serotypes and these” to “emerging serotypes, and these”

Done. See line 36

Lines 41-42: Based on the responses to reviewers, I am assuming the “0-347” refers to pairwise SNP differences calculated between strains; if this is correct, I would thus re-word to something like “The results indicate that the strains encompass 853 polymorphic sites (core-SNPs), with isolates differing from one another by 0-347 core SNPs”, as “per strain” here is ambiguous.

This was changed. See lines 41-42

Lines 191-192: Which R functions/packages were used to perform MDS?

We used base R v4.0.3 functions through RStudio v1.3.1093 and the R packages stats v4.0.3 and dplyr v1.0.7 (with the dist, cmdscale and as_tibble functions). Visualization was made with ggplot2 v3.3.5. This information has been included in the reviewed manuscript (see lines 192-194): “using R 4.0.3 [64] with the packages stats v4.0.3 and dplyr v1.0.7 (functions: dist, cmdscale and as_tibble); visualization was generated with the ggplot2 v3.3.5 [65]

Line 197: Change “with pHeatMap” to “with the pHeatMap”

Done. See line 198

Lines 199: I would suggest changing “the changes in the genome core-positions” to “core SNPs” for conciseness/readability, as the original sentence sounds strange/is difficult to follow.

Done. See line 200

Line 201: In the previous version of the manuscript, Gubbins was used to remove recombination prior to running snp-sites; why was this step omitted this time?

Gubbins was mistakenly used in the previous version. This was pointed out by the reviewer, and it was also suggested not to use, thus we decided to follow this suggestion. As this is a method used for removing recombination, only intended to work with intra-lineage recombination.

Lines 201-202: I’m assuming core SNPs were extracted usng snp-sites; if this is the case, I would change “These polymorphic positions” to “Core SNPs”, and specify that the “-c” option was used with snp-sites (i.e., the option which outputs a core SNP alignment).

This was clarified in the revised version (See lines 201-202).

Lines 202-203: Many more details need to be provided regarding how MrBayes was used here, as it is not replicable as-is; e.g., I’m assuming core SNPs produced by Snippy/snp-sites were used as input; was an ascertainment bias correction applied to account for the use of solely variant sites, and if so, how was this performed? Which model was specified? Which priors were used?

Yes, the core SNPs alignment (produced by Snippy/snp-sites) were used as input. For that version the GTR was used as the model of substitution and Gamma as the model for among-site rate variation. Nonetheless, for this new revised version we decided to change to ML as there was no independent information to parameterize priors in a Bayesian analysis and also there were no corrections for ascertainment bias implemented.

Hence, for a new versión of the analysis RAxML was used with Jukes-Cantor model and Lewis ascertainment bias correction, 100,000 bootstrap and 100 searches for the best tree (as discussed in Leaché et al., 2015). Information regarding this has been added to the reviewed manuscript (see lines 202-207): “Next, a Maximum Likelihood phylogeny was reconstructed using the core SNPs alignment as input through the implementation of RAxML v8 [71]; with Jukes-Cantor model and Lewis ascertainment bias correction (as discussed in [72] for SNPs phylogenies). Also, 100,000 bootstrap and 100 searches for the best tree were performed ("-m ASC_GTRGAMMA", "--JC69", "--asc-corr=lewis" options).”

Reference: Leaché, A. D., Banbury, B. L., Felsenstein, J., De Oca, A. N. M., & Stamatakis, A. (2015). Short tree, long tree, right tree, wrong tree: new acquisition bias corrections for inferring SNP phylogenies. Systematic biology, 64(6), 1032-1047.

Lines 205-206: I would suggest moving this sentence up to where Snippy is discussed, as the reference genome is used with Snippy/relevant for SNP calling.

This was included and the overall section was reworded. See lines 327-331: “We evaluated a phylogeny constructed using core SNPs and calculated the distances between the genomes using the identified core-SNPs (meaning the SNPs identified in all the analyzed strains) to determine differences as well. The SNPs were identified against the reference genome of Salmonella Infantis CVM-N17S1509 (GenBank: CP052817.1)”

Line 209: Change “with pHeatMap” to “with the pHeatMap”

Done. See line 212

Line 323: Consider changing “SNPs phylogeny” to “a phylogeny constructed using core SNPs”

Done. See line 327-328

Lines 331-333: What evidence is there to support the claim that slaughterhouse conditions are more likely to favor more diverse strains (e.g., how can one rule out that a strain, which is not adapted to the slaughterhouse environment, was introduced recently into the slaughterhouse from an external location/source?)?

The whole statement was changed, and focus was shifted to explain causes for the differences found. See lines 337-340: “However, we cannot conclude if these differences are due to natural selection with the disinfection protocols acting as selector or if this structure is maintained as the consequence of gene flow associated with movement of personnel or materials.”

Lines 359-360: How were associations determined (or, in this case, the lack thereof) here?  

This phrased was reworded to better explain the idea. See lines 367-370

Lines 406-408: Because the genomes here are draft genomes, one cannot prove definitively that the IncFIB plasmid harbored the AMR and/or virulence genes; was the IncFIB replicon colocalized with the AMR/virulence factors (i.e., were these elements all on the same contig)? This may offer more proof to support this statement.

Colocalization of plasmid replicons with the interest genes (AMR or virulence) was not addressed in our work. The suggestions we made are based on prior knowledge, references, and associations.

Lines 409-410: What evidence is there that these virulence factor determinants are on the specified plasmids, given that the genomes are not closed genomes (see my comment for 406-408 above)?

Indeed, we cannot conclude that the genes or factors of interest (AMR or virulence) are present actually in these plasmids as we are working with draft genomes. To evaluate colocalization would be one approximation. Nonetheless, here we are basing the suggestions on previous descriptions of these plasmids.

Line 446: Consider changing “SNPs-based” to “SNP-based”, as it sounds better.

Done. See line 460

Line 447-448: What was the rationale for rooting the tree using the reference genome? Based on the midpoint rooted phylogeny (Figure 4) and core SNP distance matrix, the reference genome doesn’t appear to be an outgroup.

Yes, the reference strain was not an outgroup (as previous Figure 4 shows). The decision to root the tree with the reference strain for figure 6 was purely cosmetic, to focus the attention on the studied strains. Nonetheless, this tree was replaced for the one produced with a new analysis and no outgroup nor rooting was used (see new Fig 6).

Lines 445-454: What are the branch length units/tree scale? Additionally, what do the node labels of the phylogeny denote?  

The branch length units/tree scale indicate the number of substitutions per site and node labels are bootstrap values. Nonetheless, this tree was replaced for the one produced with a new analysis and no scale nor supporting values are displayed.

Lines 457-461: This is confusing, as above (Lines 431-437) it is mentioned that there is “an apparent random distribution of the strains” and that “strong associations or clustering based on genomic variability and source of the sample within the production line”…”could not be detected”.

This statement was reworded to better explain the idea and reduce confusing. See lines 440-446: “There is no genetic structure within the sequenced S. Infantis among the sampled sites, the processes that are responsible for this pattern might be associated with founder effect, if all strains come from the same unsampled source. Another factor in play might be natural selection, as the facility might be seeded with different strains, but the disinfection protocols select for similar strains. Moreover, gene flow within the facility might be associated with movement of personnel and/or contaminated materials that might contribute to gene horizontal transfer or even direct contamination”

Line 484: Change “resistance bacteria” to “resistant bacteria”

Done. See line 497

Line 488: It is unclear why mealworms are discussed here

This was removed, as it did not add to the main discussion. And it was requested by other reviewers to remove unnecessary references.

Supplementary Figures: The Supplementary Figures would benefit from figure legends.

Legends for the supplementary figures S1 and S2 are in the lines 587-590, at the end of the manuscripts (as established by the journal format).

Reviewer 2 Report

It is more better to decrease cited references

Author Response

We tried to reduce as many citations as possible as requested, however, the new analyzes and comparisons requested by the other reviewers did not allow us to further reduce the number of references, since they are pertinent to what has been described and otherwise it would lose the intended sense.